# Single-cell RNA sequencing of the *Strongylocentrotus purpuratus* larva reveals the blueprint of major cell types and nervous system of a non-chordate deuterostome

Periklis Paganos[1], Danila Voronov[1], Jacob M Musser[2], Detlev Arendt[2], Maria Ina Arnone[1]*

[1]Stazione Zoologica Anton Dohrn, Department of Biology and Evolution of Marine Organisms, Naples, Italy; [2]European Molecular Biology Laboratory, Developmental Biology Unit, Heidelberg, Germany

**\*For correspondence:**
miarnone@szn.it

**Competing interest:** The authors declare that no competing interests exist.

**Abstract** Identifying the molecular fingerprint of organismal cell types is key for understanding their function and evolution. Here, we use single-cell RNA sequencing (scRNA-seq) to survey the cell types of the sea urchin early pluteus larva, representing an important developmental transition from non-feeding to feeding larva. We identify 21 distinct cell clusters, representing cells of the digestive, skeletal, immune, and nervous systems. Further subclustering of these reveal a highly detailed portrait of cell diversity across the larva, including the identification of neuronal cell types. We then validate important gene regulatory networks driving sea urchin development and reveal new domains of activity within the larval body. Focusing on neurons that co-express *Pdx-1* and *Brn1/2/4*, we identify an unprecedented number of genes shared by this population of neurons in sea urchin and vertebrate endocrine pancreatic cells. Using differential expression results from Pdx-1 knockdown experiments, we show that Pdx1 is necessary for the acquisition of the neuronal identity of these cells. We hypothesize that a network similar to the one orchestrated by Pdx1 in the sea urchin neurons was active in an ancestral cell type and then inherited by neuronal and pancreatic developmental lineages in sea urchins and vertebrates.

## Editor's evaluation

This work provides a comprehensive analysis of cell state specification of a whole deuterostome organism, the sea urchin *Strongylocentrotus purpuratus*. It is also vigorous example for the use of single-cell sequencing to identify cell type homologies across evolution. The paper is thus of significant interest to scientists within the broad fields of developmental biology and evolution, as well as to the more specific communities of researchers that use the sea urchin as a model system or those interested in employing the single-cell mRNA-sequencing technology for "non-conventional" (and marine) molecular model systems.

## Introduction

Multicellular organisms consist of numerous cell types, specialized in performing different tasks that guide growth and survival. During embryonic development, cells go through rounds of proliferation, specification and differentiation into cell types with distinct functions. The information for this

developmental diversification lies in the genome and the spatio-temporal expression of regulatory genes that specifies the molecular fingerprint of a given cell type (*Fu et al., 2017*).

The identity of each cell type is established, controlled, and maintained by distinct Gene Regulatory Networks (GRNs). GRNs are logical maps of the regulatory inputs and outputs active in a cell at a given place and time, and are enacted by transcription factors, signaling molecules and terminal differentiation genes (*Davidson et al., 2003*; *Davidson and Erwin, 2006*). GRNs have been studied in a variety of organisms ranging from plants to animals in order to analyze the gene interactions at a specific time and place during the life of an organism (*Krouk et al., 2013*), and have been used for understanding the relationship between genome, development and evolution (*Davidson and Erwin, 2006*). Therefore, understanding the genetic mechanisms that provide cell types with a specific identity, and the conservation of this identity across animal taxa, is essential for understanding cell type function and evolutionary history (*Arendt, 2008*; *Arnone et al., 2016*).

Until recently, most approaches for comparing cell types relied on the description of distinct morphological features, that is linked to the functionality of a given cell type, the identification of molecular markers, perturbation of gene expression and fate mapping. However, technological advances in microfluidics and nucleic acid barcoding now allow the high-throughput recognition of an organism's cell types at a single-cell level. In particular, single-cell RNA sequencing (scRNA-seq) technology, developed during the last decade, is a powerful method used to unravel the transcriptional content of individual cells, resulting in the identification of distinct cell types in an unbiased manner (*Tang et al., 2009*; ; *Klein et al., 2015*). ScRNA-seq involves dissociation of an organism, organ, or tissue into single cells, isolation and capture of the single cells into droplets, specific barcoding of individual mRNAs, and sequencing of transcriptomic content of each cell. Computational analysis can then identify putative cell type families by clustering cells with similar transcriptional profiles. Further analysis of such cell clusters can lead to the identification of distinct cell types.

Within deuterostomes, echinoderms are a member of the phylogenetic sister group to chordates, making them an ideal model for understanding the origin and diversification of deuterostome and chordate cell types (*Arnone et al., 2015*). Sea urchin embryos and larvae have also been extensively used to unravel the general mechanisms of cell type specification and differentiation during development (*Cameron and Davidson, 1991*; *Davidson et al., 1998*; *McClay, 2011*; *Lyons et al., 2012*; *McClay et al., 2020*). The main reason for this lies in the ease with which different cell types and biological processes can be observed in the optically transparent embryos and larvae. Among the most well-studied sea urchin cell types are those comprising the nervous (*Bisgrove and Burke, 1987Burke et al., 2006a*, *McClay et al., 2018*), immune (*Rast et al., 2006*; *Ho et al., 2017*) and digestive systems (*Annunziata et al., 2014*; *Annunziata and Arnone, 2014*; *Perillo and Arnone, 2014*; *Perillo et al., 2016*), and of both musculature (*Andrikou et al., 2013*; *Andrikou et al., 2015*) and skeleton (*Okazaki, 1965*; *Duloquin et al., 2007*; *Rafiq et al., 2012*; *Sun and Ettensohn, 2017*). For these, the developmental origins and gene regulatory wiring has been described in great detail, making the sea urchin an ideal model for GRN comparative analyses in development and evolution (*Cary et al., 2020*).

Here, we take advantage of the detailed characterization of the sea urchin cell types performed over the years, the available cell-type-specific molecular markers, and the ease with which the sea urchin larvae are dissociated into single cells, to perform scRNA-seq and generate a comprehensive atlas of sea urchin larval cell type families. Our findings suggest that the larva consists of 21 genetically distinct cell clusters, representing distinct cell type families (*Shekhar and Menon, 2019*), which we validate using fluorescent in situ hybridization (FISH) and immunohistochemistry (IHC). Based on previous studies tracing developmental lineage, we assign cell type families to the specific embryonic germ layers they derived from. Comparing the transcription factor content of the grouped cell clusters reveals that most transcription factors are expressed pleiotropically independently of their germ layer origins, yet tend to be cell type family-specific within a germ layer. In addition, we illustrate how single-cell data complement and validate previously studied GRNs, and also reveal novel cellular domains where these GRNs are likely also activated. Lastly, we investigate neuronal diversity in the sea urchin larva, identifying 12 distinct neuronal cell types. Among these, we recover a unique neurosecretory type controlled by *Sp-Pdx1* and *Sp-Brn1/2/4* exhibiting a pancreatic-like gene expression signature (*Perillo et al., 2018*). Our results confirm and extend this pancreatic-like signature, suggesting that an ancestral neuron in early deuterostomes may have given rise to the endocrine cells in the

vertebrate pancreas. Supporting this, knockdown of *Sp-Pdx1* shows it is necessary for differentiation of this pancreatic-like neuronal endocrine population, indicating it has an evolutionary conserved role as a mediator of endocrine fate.

## Results

### Building a cell type atlas of the sea urchin larva with single-cell transcriptomics

Sea urchin early pluteus larvae were cultured and collected at three dpf. We performed single-cell RNA sequencing on six samples from four independent biological replicates. Individual samples were dissociated into single cells using a gentle enzyme-free dissociation protocol and using the 10 x Chromium scRNA-seq system (*Figure 1A*). In total, transcriptomes from 19,699 cells were included in the final analysis. To identify sea urchin larval cell types, we used Louvain graph clustering as implemented in the Seurat pipeline (see Materials and methods). This revealed 21 genetically distinct cell clusters (*Figure 1B*, *Figure 1—figure supplement 1A,B*), each representing an individual cell type or a set of closely related cell types in the early pluteus larva.

Next, we set out to explore the identity of our initial 21 cell clusters. We first assigned preliminary identities to each cluster based on the expression of previously described cell type markers, benefiting from the unique and rich knowledge on sea urchin developmental lineages: ciliary band (*Btub2*) (*Harlow and Nemer, 1987*), apical plate (*Hbn*) (*Burke et al., 2006a*), aboral ectoderm (*Spec2a*) (*Yuh et al., 2001*), lower oral ectoderm (*Bra*) (*Wei et al., 2012*), upper oral ectoderm (*Gsc*) (*Wei et al., 2012*), neurons (*SynB*) (*Burke et al., 2006a*), esophageal muscles (*Mhc*) (*Andrikou et al., 2013*), coelomic pouches (*Nan2*) (*Juliano et al., 2010*), blastocoelar cells (185/333) (*Ho et al., 2017*), immune cells (*Gcm*) (*Materna et al., 2013*), skeleton (*Msp130*) (*Harkey et al., 1992*), anus (*Hox11/13b*), intestine (*Cdx*), pyloric sphincter (*Pdx-1*), different stomach domains (*Chp, ManrC1a, Endo16*) (*Annunziata and Arnone, 2014*; *Annunziata et al., 2019*), exocrine pancreas-like domain (*Ptf1a*) (*Perillo et al., 2016*), cardiac sphincter (*Trop1*) (*Yaguchi et al., 2017*), and esophagus (*Brn1/2/4*) (*Cole and Arnone, 2009*). Further, we grouped the cell clusters according to their embryonic germ layer origin (*Figure 1C*) using knowledge from previous lineage tracing experiments (*Angerer and Davidson, 1984*; *Cameron et al., 1987*).

To validate these identities, we identified all genes expressed in each cell type, totaling in 15,578 WHL genes (transcriptome models) and 12,924 genes (SPU gene models) (*Supplementary file 1*), and performed in situ hybridization on a selected set of these together in combination with previously described markers. Based on this, we mapped 20 of the 21 clusters to distinct larval domains and confirmed their identity (*Figure 1E*). Notably, the resulting expression patterns validated the initial predictions (*Figure 2—figure supplement 2*), verifying the high quality of the single-cell dataset. Importantly, this approach identified various new markers for each cell type family, including *Sp-FbsL_2* (ciliary band; *Figure 2A1*), *Sp-hypp_2386* (skeletal cells; *Figure 2A9*), and *Sp-Serp2/3* (exocrine pancreas-like cells; *Figure 2A14*). The 21st cluster, which had a poorly-defined molecular signature as judged by the low number and expression level of total and marker genes combined with the lack of specific localization, likely represents not fully differentiated cells (*Figure 1—figure supplement 1D,F*) and we refer to this cluster as undefined.

Our scRNA-seq analysis and in situ hybridization protocol unraveled novel expression domains for several previously described cell type markers. For instance, the transcription factors *Sp-SoxC* and *Sp-Hbn*, previously described in early neuronal specification (*Garner et al., 2016*; *Wei et al., 2016*; *Yaguchi et al., 2016*), were predicted by our scRNA-seq analysis to also be expressed in skeletal cells (*Figure 2B*). Likewise, the FGF signaling ligand, *Sp-Fgf9/16/20*, is known to be involved in skeletal formation and is expressed in specific populations of PMCs (*Adomako-Ankomah and Ettensohn, 2014*). ScRNA-seq indicates it is also expressed in oral ectoderm, cardiac sphincter, intestine, and anus (*Figure 2B*). Using in situ hybridization and immunohistochemical detection methods we confirmed the predictions and we found *Sp-SoxC* expressed in *Sp-Fgf9/16/20* positive PMCs and *Sp-Hbn* expressed in skeletal cells located in the vertex (*Figure 2* 2D1-4 & 2D5-9). Moreover, we found *Sp-Fgf9/16/20* transcripts localized in distinct gut domains corresponding to the cardiac sphincter, intestinal, and anal regions (*Figure 2*).

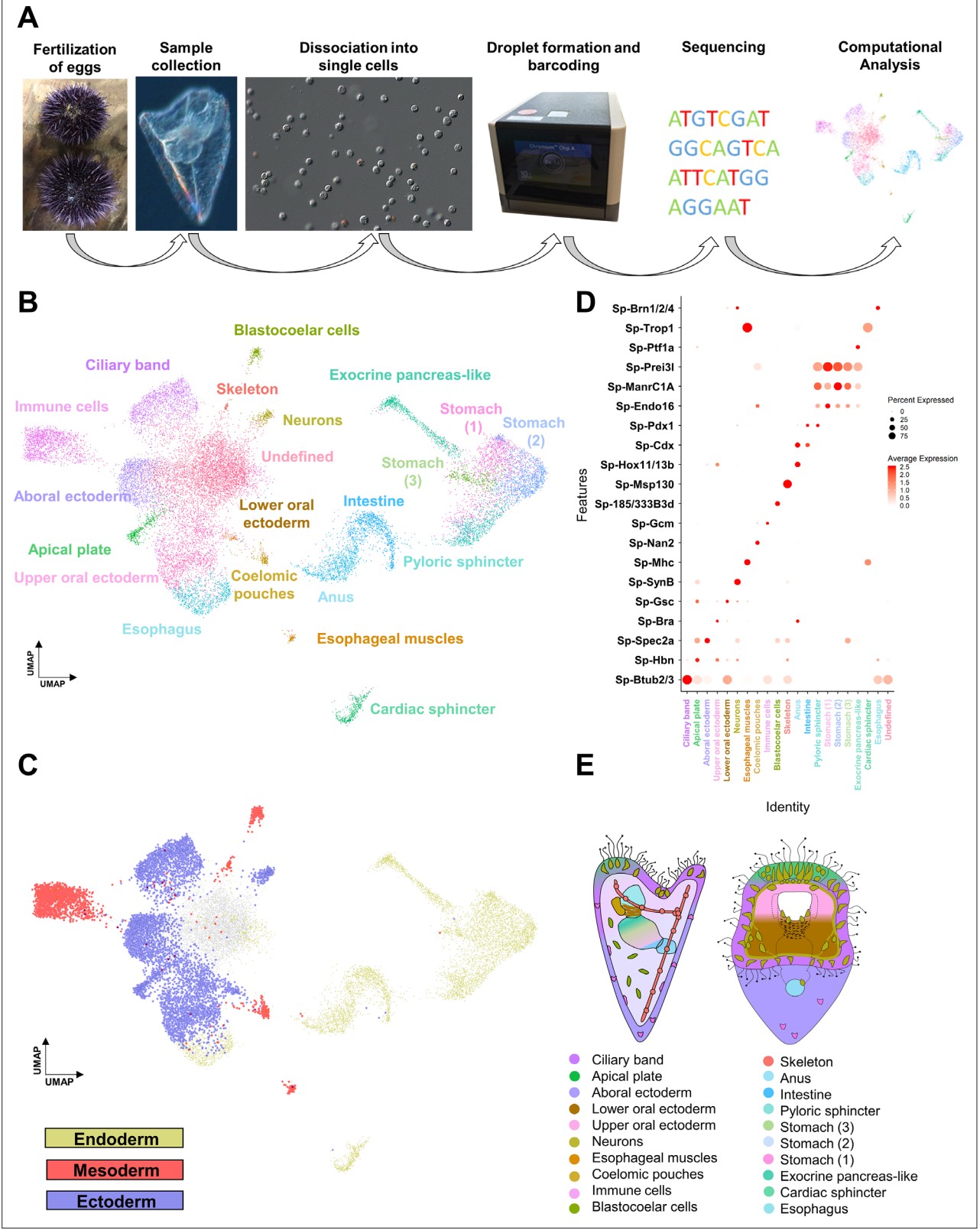

**Figure 1.** Cell type family atlas of the three dpf *S. purpuratus* larva. (**A**) Single-cell RNA sequencing pipeline from gamete fertilization to computational analysis. (**B**) UMAP showing three dpf larval cells colored by their assignment to the initial set of 21 distinct cell clusters. (**C**) UMAP with cells colored by germ layer of origin: endoderm (yellow), mesoderm (red), and ectoderm (blue). (**D**) Dotplot of gene markers specific to cell clusters. (**E**) Illustration depicting location of cell type families on different larval domains. Color-code is the same as in (**B**).

*Figure 1 continued on next page*

*Figure 1 continued*

The online version of this article includes the following figure supplement(s) for figure 1:

**Figure supplement 1.** Overlap of the different replicates and characterization of the undefined cluster.

We compared the limits of detection by in situ hybridization versus single cell RNA sequencing, using the coelomic pouch cell cluster as a case study. The coelomic pouches are derived from the mesoderm and the left coelomic pouch contributes to the formation of the rudiment and juvenile sea urchin after metamorphosis (*Strathmann, 1987*; *Smith et al., 2008*). The formation of the coelomic pouches is complex, and includes contributions from the small micromeres, a mesodermal cell population that is set aside during early development (*Pehrson and Cohen, 1986*; *Strathmann, 1987*). Previous attempts to characterize this population had involved screening of genes active in germline determination and maintenance in other species and revealed that, while some germ-line-specific transcripts and proteins were found exclusively expressed in the small micromeres and the coelomic pouch of the sea urchin embryo (*Juliano et al., 2006*), the majority of the genes tested by in situ hybridization were not enriched in this cell type. Interestingly, plotting the Juliano and co-authors' gene list, alongside previously described coelomic pouch specific genes (*Luo and Su, 2012*; *Martik and McClay, 2015*), they were all found in our analysis to be expressed in the same cell cluster (*Figure 2—figure supplement 3*). This suggests a higher sensitivity of single-cell RNA sequencing compared to the in situ hybridization, adding crucial missing information on the molecular fingerprint of a complex cell type.

Lastly, to determine which cells in the larva were undergoing active proliferation, we plotted expression of cell division markers in sea urchin, including *pcna*, DNA polymerases, DNA ligases, condensins, and centromere proteins (*Perillo et al., 2020*). The majority of cell proliferation genes were found to be enriched in the ciliary band, apical plate, coelomic pouch, immune, and skeletal cell clusters (*Figure 2—figure supplement 4A*). We also observed Cdk genes enriched in several endodermally derived cell populations (*Figure 2—figure supplement 4A*). Validating this, we observed S-phase cells in ciliary band, apical plate, oral ectoderm, endodermal and skeletal cells using Edu pulse labeling (*Figure 2—figure supplement 4B*). In contrast, we did not observe Edu fluorescence in cell populations that lacked expression of proliferation markers, such as the aboral ectoderm, suggesting that indeed they may not be proliferating.

## Shared lineage information of the larval cell type families

We next compared transcription factor profiles of early larval cell type families. In general, cell populations derived from the same germ layer shared more factors with each other than with cell populations from other germ layers (*Figure 3*, *Figure 3—figure supplement 1*), consistent with the finding that cell type expression programs often retain information about their developmental lineage (*Sladitschek et al., 2020*). Further analysis, however, revealed very few transcription factors specific to cells from a single germ layer, contrasting with many regulators expressed in derivatives of more than one germ layer (*Figure 3A*), and nearly one third (n = 187) shared by cell type families from all three layers. Unexpectedly, mesodermal cell populations share expression of more transcription factors with ectodermal than with endodermal cell populations, even though they are more closely linked to endodermal lineage in development. We also noted that cell type families derived from the same germ layer share up to one third of transcription factors, while the majority are cell type-specific (*Figure 3B, D and E*). In general, neighboring cell populations and those with common developmental origins share a larger number of TFs (*Figure 3D*), compared to cell populations with different developmental histories (*Figure 3C*).

To further characterize the regulatory profile of larval cell type families we set out to identify the expression profiles of members of major transcription factor families (*Figure 4*, *Figure 4—figure supplement 1*). In *S. purpuratus*, most homeobox transcription factors were previously found expressed at the gastrula stage (two dpf), with several members expressed in domains derived from all three germ layers (*Howard-Ashby et al., 2006*). Our single-cell analysis, although at a later developmental time point, supports these findings, and further refines our understanding of their expression to specific cell type families. In the early pluteus larva, most homeobox class transcription factors are enriched in ectodermally derived cell populations, such as the apical plate and neurons. In contrast, ANTP Class and HNF class transcription factors are enriched in endodermal derivatives (*Figure 4*).

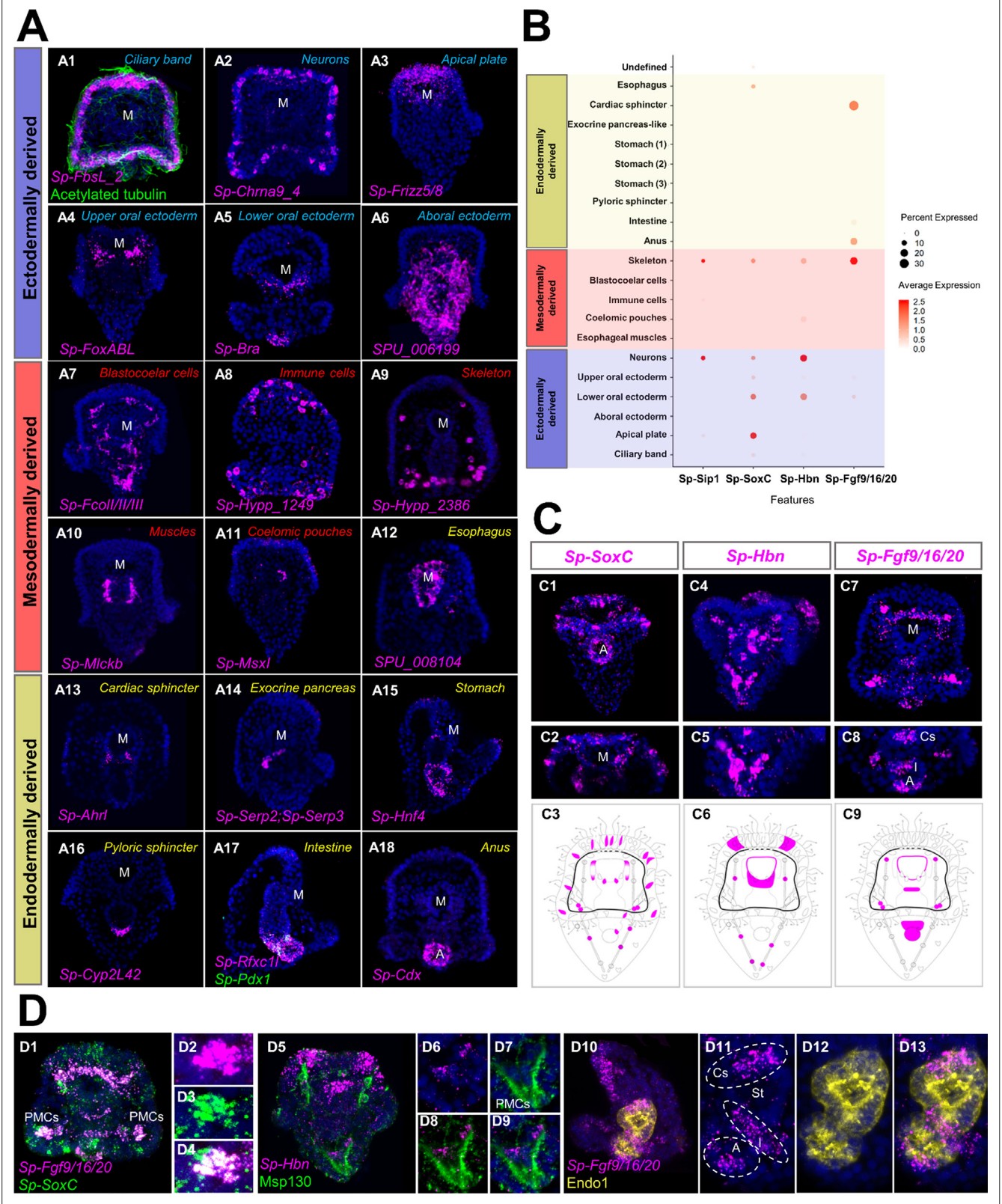

**Figure 2.** Validation of scRNA-seq predictions and novel expression domains. (**A**) FISH of *S. purpuratus* three dpf larvae with antisense probes for *Sp-FbsL_2* (**A1**), *Sp-Chrna9_4* (**A2**), *Sp-Frizz5/8* (**A3**), *Sp-FoxABL* (**A4**), *Sp-Bra* (**A5**), *SPU_006199* (**A6**), *Sp-FcoII/II/III* (**A7**), *Sp-Hypp_1249* (**A8**), *Sp-Hypp_2386* (**A9**), *Sp-Mlckb* (**A10**), *Sp-MsxI* (**A11**), *SPU_008104* (**A12**), *Sp-Ahrl* (**A13**), *Sp-Serp2; Sp-Serp3* (**A14**), *Sp-Hnf4* (**A15**), *Sp-Cyp2L42* (**A16**), *Sp-Rfxc1l* (**A17**), *Sp-Pdx1* (**A17**) and *Sp-Cdx* (**A18**). Color-code indicates germ layer embryonic origin: endoderm (yellow), mesoderm (red), ectoderm

*Figure 2 continued on next page*

*Figure 2 continued*

(blue). Immunofluorescent detection of acetylated tubulin in ciliary band (green). (**B**) Dotplot of *Sp-Sip1, Sp-SoxC, Sp-Hbn,* and *Sp-Fgf9/16/20* expression showing previously described and novel expression domains. (**C**) FISH of *S. purpuratus* three dpf larvae with antisense probes for *Sp-SoxC* (**C1–C2**), *Sp-Hbn* (**C4–C5**), and *Sp-Fgf9/16/20* (**C7–C8**). Illustrations depicting all the expression domains of *Sp-SoxC* (**C3**), *Sp-Hbn* (**C6**), and *Sp-Fgf9/16/20* (**C9**). Illustrations depicting all the expression domains of *Sp-SoxC* (**C3**), *Sp-Hbn* (**C6**), and *Sp-Fgf9/16/20* (**C9**). Nuclei are labeled with DAPI (in blue). All images are stacks of merged confocal Z sections. (**D**) FISH of *S. purpuratus* three dpf larvae with antisense probes for *Sp-SoxC* and Sp-Fgf9/16/20 (**D1–D4**), for *Sp-Hbn* combined with immunohistochemical detection for the skeletal cells marker Msp130 (**D5–D9**) and *Sp-Fgf9/16/20* with immunohistochemical detection for the midgut and hindgut protein Endo1 (**D10–D13**). A, Anus; Cs, Cardiac sphincter; I, Intestine; M, Mouth; PMCs, Primary mesenchyme cells; St, Stomach.

The online version of this article includes the following figure supplement(s) for figure 2:

**Figure supplement 1.** Expression pattern of genes used to annotate the clusters.

**Figure supplement 2.** ScRNA-seq predicted expression patterns of genes used to annotate the clusters.

**Figure supplement 3.** ScRNA-seq is able to detect expression patterns, previously undetectable by in situ hybridization.

**Figure supplement 4.** Proliferation status and dynamics of the larval cell type families.

Other major transcription factor families, such as the Forkhead, Ets, and Zinc-finger families, members of which are expressed throughout sea urchin embryogenesis (*Tu et al., 2006*; *Rizzo et al., 2006*; *Materna et al., 2006*), are also expressed across a spectrum of cell populations. Forkhead and zinc-finger transcription factors are highly expressed in specific cell type families of all three germ layer derivatives, whereas Ets family TFs are enriched in ectodermal and mesodermal derivatives (*Figure 4*).

The active regulatory state of a given cell type family is an immediate consequence of the gene regulatory network active at this time point. Previous research in sea urchin has described in detail many regulatory networks active during embryonic and larval development. Our scRNA-seq data is broadly consistent with previous studies, yet also identifies new domains and cell populations in which these regulatory networks may be active. For instance, we plotted all transcription factors known to be active in specifying coelomic pouch cells (*Martik and McClay, 2015*). Our data confirmed their co-expression in coelomic pouch, but also revealed their co-expression in the apical plate (*Figure 5A*). Similarly, plotting genes involved in the aboral ectoderm gene regulatory network (*Ben-Tabou de-Leon et al., 2013*), we found all genes in both the aboral ectoderm cluster as well as in the apical plate cells (*Figure 5B*). On the other hand, plotting members of the pre-gastrula skeletogenic mesoderm regulatory network (*Figure 5C*) revealed most were still active in the pluteus larva and specific to skeletal cells (*Oliveri et al., 2008*; *Shashikant and Ettensohn, 2019*; *Khor et al., 2019*; *Ettensohn, 2020*). These findings illustrate the immediate benefit of our dataset to drastically expand our knowledge of larval regulatory networks. Finally, our scRNA-seq recreates a nearly identical three dpf endoderm expression pattern atlas as that published previously by our group using more traditional methods (*Annunziata et al., 2014*), providing additional information on each gene's average expression level and the percentage of cells expressing each marker (*Figure 5—figure supplement 1*).

## Unravelling the neuronal diversity and molecular signature of the nervous system

The assessment of neuronal cell type diversity is an important step for unravelling the evolution and function of the nervous system. The sea urchin free swimming larva is equipped with a nervous system consisting of interconnected ganglia (*Burke et al., 2006a*) that allows the animal to respond to environmental stimuli and coordinate its swimming behavior (*Soliman, 1983*; *Katow et al., 2010*). Several neuronal types, including apical and ciliary band neurons, as well as neurons along the digestive tube, have been previously identified and their specification described in detail (*Burke et al., 2006a*, *Burke et al., 2006b*, *Wei et al., 2009*; *Wei et al., 2011*; *Burke et al., 2014*; *Garner et al., 2016*; *Wei et al., 2016*; *McClay et al., 2018*; *Perillo et al., 2018*; *Wood et al., 2018*).

Our initial clustering analysis resolved single clusters for neuronal cells, as well as for PMCs and immune cells. However, expression of known markers suggested the presence of distinct subclusters in each of these cell type family groups. In order to investigate this, we independently performed subclustering and re-analysis of the neuronal, immune, and PMC cells. Subclustering of each of these initial major clusters revealed 12 neuronal, 8 immune, and 5 PMC subclusters, each likely representing distinct cell types (*Figure 6*, *Figure 6—figure supplements 1 and 2*). Two of the immune subclusters

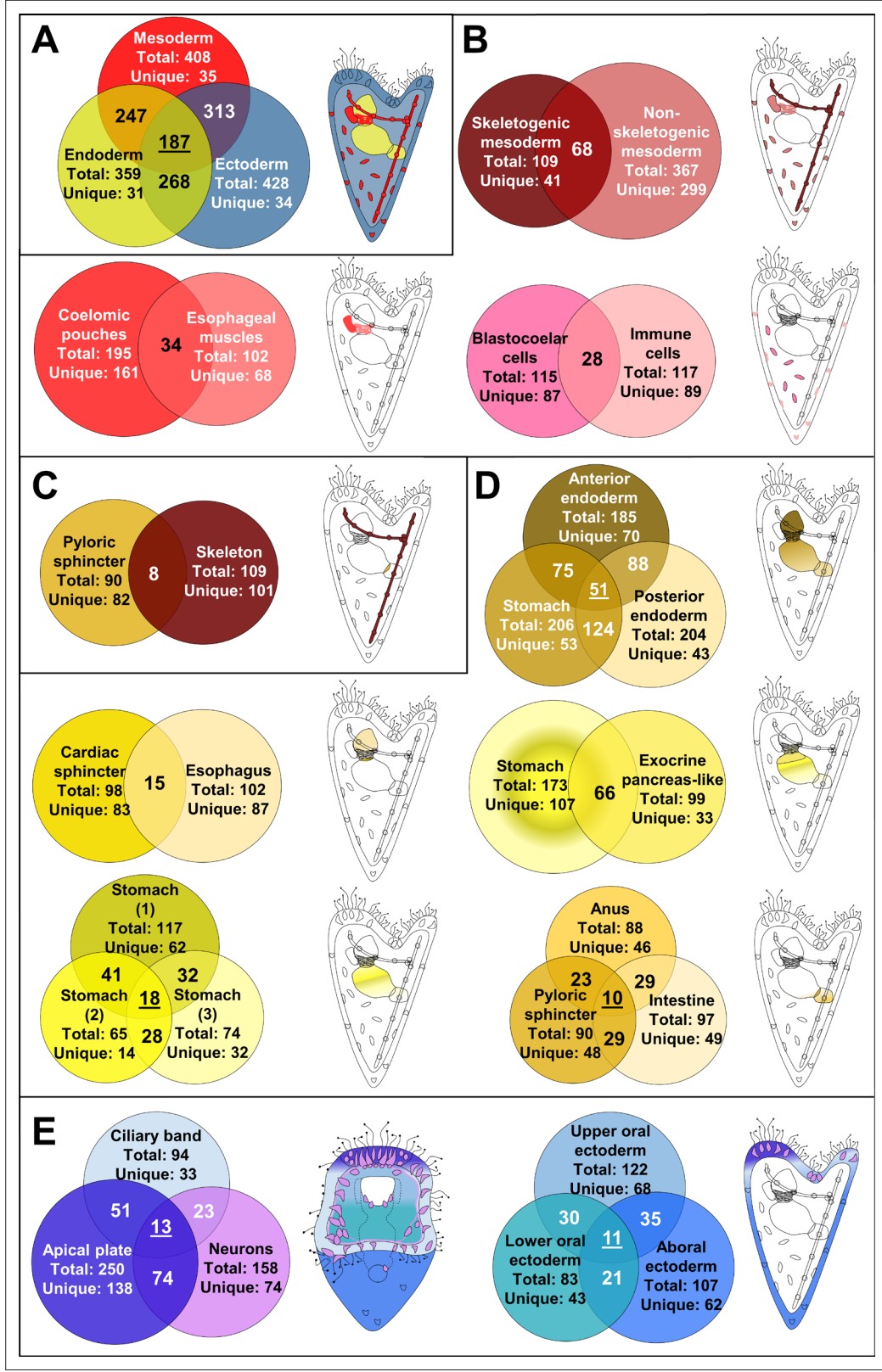

**Figure 3.** Regulatory states of the three dpf *S. purpuratus* larva. (**A**) Comparison of the transcription factor content per germ layer. Venn diagram showing the shared and unique transcription factors per germ layer. Ectodermally derived cell type families are shown in blue, mesodermally derived in red, and endodermally derived in yellow. (**B**) Comparison of the transcription factor content across mesodermal lineages and cell type families. Venn diagram

*Figure 3 continued on next page*

*Figure 3 continued*

showing the shared and unique transcription factors per comparison. (**C**) Transcription factor content comparison of pyloric sphincter (endodermally derived) and skeletal cells (mesodermally derived), used as a negative control of our comparison. (**D**) Comparison of the transcription factor content per endodermal lineage and endodermally derived cell type families. Venn diagram showing the shared and unique transcription factors per comparison. (**E**) TF signature comparison of ectodermally derived cell type families. Venn diagram showing the shared and unique transcription factors per comparison. Cartoons indicated the relative position of each cell type family/lineage. Mesodermal cell type families/lineages are shown in shades of red, endodermal ones in shades of yellow and endodermal ones in shades of blue.

The online version of this article includes the following figure supplement(s) for figure 3:

**Figure supplement 1.** Cell type family trees of the three dpf pluteus larva.

expressed polyketide synthase 1 (*Sp-Pks1*), suggesting these represent sea urchin pigment cell populations (***Calestani and Rogers, 2010***). We also found a subcluster of immune cells that expresses the membrane attack complex/perforin family gene (*Sp-MacpfA2*), suggesting this corresponds to immune system globular cells (***Figure 6—figure supplement 2***). Notably, our finding of 5 PMC subclusters corroborates previous reports showing five distinct groups of PMC cells along the syncytium (***Sun and Ettensohn, 2014***; ***Figure 6—figure supplement 1***).

To annotate the 12 neuronal cell types revealed via subclustering, we took advantage of the extensive previous work investigating neurogenesis and neuronal differentiation in sea urchin. Plotting neuronal markers, we resolved unique molecular signatures for each subcluster and assigned each a putative identity and location in the larva (***Figure 6B–D***). To validate this, we then conducted in situ hybridization experiments for gene markers labeling these specific neuronal populations (***Figure 6D***), including genes encoding transcription factors (*SoxC*, *Delta*, *Ngn*, *Prox1*, *Isl*, *Hbn*, *SoxB2*, *Otx*, *NeuroD1*, *Six3*), as well as members of neurotransmitter (*Ddc*, *Nacha6*, *Chrna9_4*, *Tph*), and neuropeptidergic signaling pathways (*An*, *Salmfap*, *Trh*).

The sea urchin larva neuronal differentiation proceeds stepwise, including transient expression of the Notch ligand Delta, followed by expression of the transcription factors *SoxC* and *Brn1/2/4* (***Garner et al., 2016***). During the final stages of neurogenesis, the transcription factors *Sip1*, *Z167*, *Ngn,* and *Otp* regulate differentiation of diverse neuronal populations, including apical and ciliary band neurons (***Wei et al., 2016***; ***McClay et al., 2018***). In our data, we observed *Sp-Delta*, *Sp-SoxC*, and *Sp-Brn1/2/4,* as well different combinations of the transcription factors mentioned above, co-localize in three neuronal populations (subclusters 1, 2, and 4), indicating neuronal differentiation is taking place in those three subclusters (***Figure 6C***). Interestingly, in one of these populations (subcluster 2) we found expression of the transcription factors *Sp-Rx*, *Sp-Hbn* (***Figure 6D10***) and *Sp-Six3* (***Figure 6D16***), which are known to be expressed in the periphery of the larva's apical domain (***Burke et al., 2006a***, ***Wei et al., 2009***). This suggests that this population is located in the periphery of the apical plate and not within the apical organ. In the apical domain, we also detected a subcluster (number 6), which coexpress *Sp-Trh* and *Sp-Salmfap* neuropeptides (***Wood et al., 2018***), as well as *Sp-Kp* (Kissepeptin) (***Figure 6D17***).

In total, we identified three neuronal subclusters located in the apical domain (subclusters 2, 3 and 4) of the larva that express Tryptophan hydroxylase (*Tph*), which encodes a key enzyme in the serotonin biosynthesis pathway, suggesting these represent serotonergic neurons in the larva. Within the ciliary band, which comprises the larva's peripheral nervous system (***Slota et al., 2020***), we identified two distinct cholinergic subclusters (7 and 8) expressing the enzyme involved in acetylcholine biosynthesis (*Sp-Chat*) (***Figure 6C***), one of which (subcluster 8) expresses also two nicotinic acetylcholine receptors (*Nacha6*, *Chrna9*). Moreover, we identified a neuronal subcluster in close proximity with the ciliary band, which corresponds to the lateral and post-oral neurons (subcluster 9). This population has been previously characterized by our group and was found to co-express *Sp-Pdx1*, *Sp-Brn1/2/4*, and the neuropeptide Sp-An (***Perillo et al., 2018***). Using gene markers that mark differentiated neurons expressed in the rim of the larva's mouth, including *Sp-Nkx3.2* (***Wei et al., 2011***), *Sp-Isl* (***Perillo et al., 2018***), the neuropeptide *Sp-Salmfap* (***Wood et al., 2018***), and the enzyme *tyrosine hydroxylase* (*Sp-Th*) involved in the dopaminergic pathway, we identified two distinct mouth neurons subtypes (subclusters 10 and 11; ***Figure 6C***). Lastly, we found one neuronal population that, based on its molecular signature, and in particular the co-presence of *Sp-Fgfr1* and *Sp-Fgf9/16/20* (***Figure 6—figure***

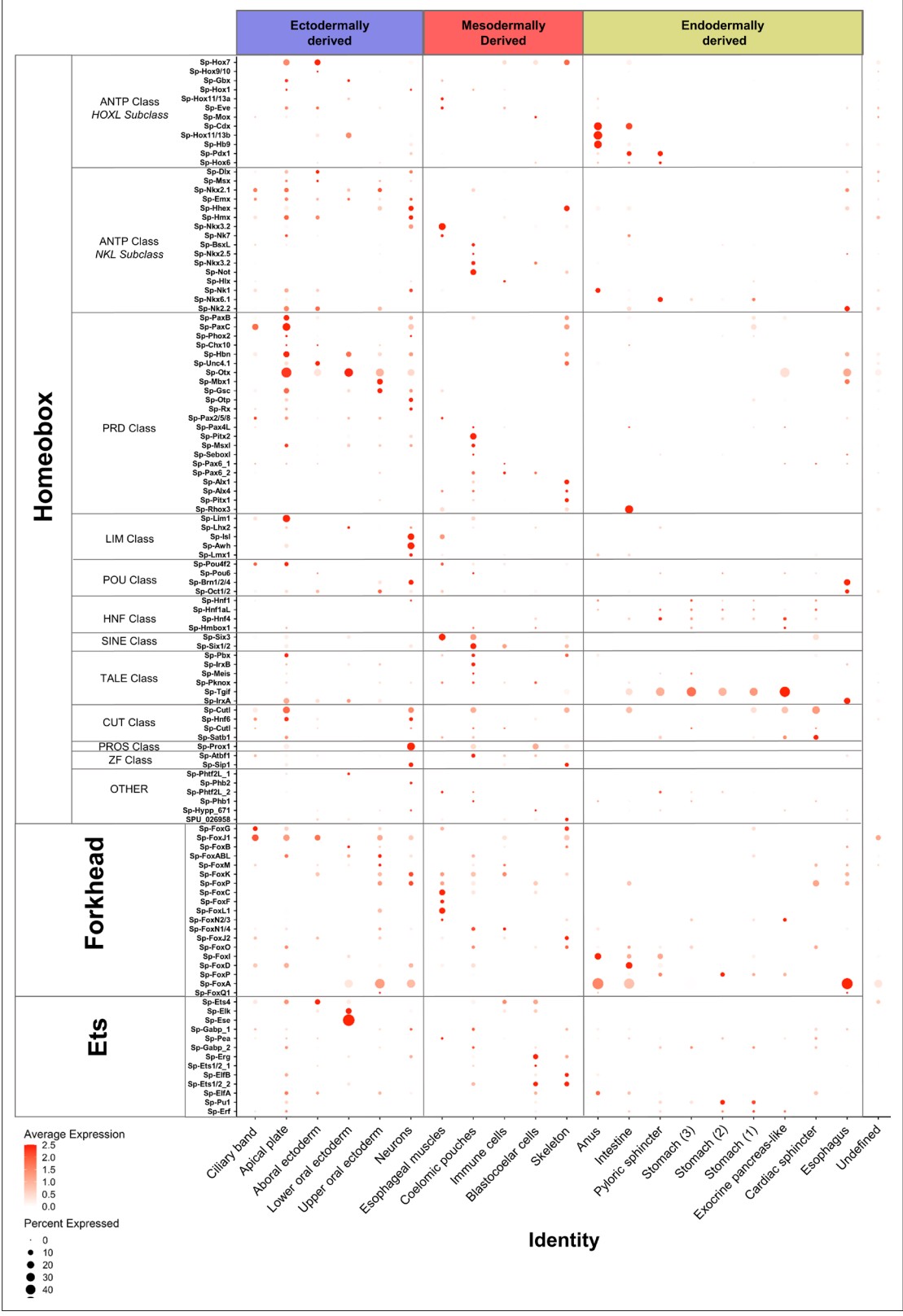

**Figure 4.** Localization of major transcription factor family members. Dotplot showing the average scaled expression of members of the Homeobox, Forkhead and Ets transcription factor families. The developmental origins of each cell type family are shown in blue for ectodermally derived, red for mesodermally derived and yellow for endodermally derived ones.

*Figure 4 continued on next page*

*Figure 4 continued*

The online version of this article includes the following figure supplement(s) for figure 4:

**Figure supplement 1.** Distribution of the Zinc finger TF family members across the cell type families.

**Figure supplement 2.** Differentially expressed TFs across cell type families.

*supplement 3*), could correspond to neurons associated with the cardiac sphincter within the esophageal endoderm (subcluster 12). Overall, our subclustering analysis shows the different neuronal subtypes present at this developmental stage at an increased resolution, describing new neuronal subtypes and providing novel markers and gene candidates for future studies of these cell types and their gene regulatory networks.

## Characterizing a neuroendocrine neuronal population controlled by Pdx-1

Previous studies from our group suggested that the nervous system of the sea urchin larva displays a strong pre-pancreatic signature, with neurons expressing genes that are involved in endocrine cell differentiation in the vertebrate pancreas (*Perillo et al., 2018*). To investigate this, we focused on the post-oral and lateral neuron subcluster, which co-expresses *Sp-Pdx1*, *Sp-Brn1/2/4*, and *Sp-An* (*Figure 6*). Double immunohistochemical staining of the neuronal marker 1E11 and Sp-An shows that these neurons lie in close proximity to the ciliary band, and project axons toward both the apical plate and ciliary band (*Figure 7A1-A3*). Double FISH of *Sp-An* and *Sp-FbsL_2*, a ciliary band marker revealed by this study, as well as double IHC of anti-Sp-An and anti-acetylated tubulin (labeling cilia), further indicate their distribution relative to the ciliary band (*Figure 7 A4–A6*). Moreover, we found that the post-oral Sp-An neurons are found in close proximity to cells of both ciliary band subclusters (*Figure 7 A7–A9*), and project axons to the cell bodies of the *Sp-Prox1* positive neurons (*Figure 7 A9*). We also observed close proximity between An positive neurons in the post oral arms with the cells of the ventral-lateral cluster of PMCs (*Figure 7 A10*), and with immune globular cells (*Figure 7 A11*). Next, we set out to validate whether novel genes predicted by our single-cell analysis to be expressed in this neuronal population can be validated in vivo. Among the genes predicted to be expressed in this population are the transcription factors *Sp-Nk1* and *Sp-Otp,* as well as the catecholaminergic and cholinergic neuronal markers Sp-Th and Sp-Chat (*Slota and McClay, 2018*), respectively. Double fluorescent in situ hybridization of *Sp-Nk1* and the neuropeptide *Sp-An*, as well as fluorescent in situ hybridization of *Sp-Otp* combined with the immunohistochemical detection of Sp-An, reveal co-localization of these three genes in the post-oral and lateral neuronal population (*Figure 7 A13–A14*), verifying the single-cell data. Additionally, double immunostainings of the anti-Sp-An with anti-Th (*Figure 7A9*) and anti-Chat antibodies suggest that these two key enzymes, involved in different neurotransmitter biosynthesis pathways, are co-produced in the Sp-*Pdx1/Sp-Brn1/2/4* neurons (*Figure 7A12,A15*).

The vertebrate orthologue of *Sp-Pdx1* is essential for pancreas development, β-cell differentiation, and maintaining mature β-cell function (*Kaneto et al., 2008*). As previously described by our group, knockdown of the pancreatic transcription factor Sp-Pdx1 results in severe downregulation of the Sp-An neuropeptide, compromising the neuroendocrine fate of this neuronal type (*Perillo et al., 2018*). To further characterize the *Sp-Pdx1/Sp-Brn1/2/4* neuronal population, we performed a comprehensive analysis of genes involved in pancreatic development and β-cell differentiation, as well as gene markers related to neuroendocrine fate. We identified a total of 20 genes, all involved in the formation and proper function of vertebrate endocrine pancreas, which are differentially enriched in the *Sp-Pdx1/Sp-Brn1/2/4* neurons (*Figure 7B*). Next, we intersected our scRNA-seq with bulk RNA sequencing data derived from Pdx1 morphants assayed at the same developmental stage in a previous study (*Annunziata and Arnone, 2014*). By coupling knowledge of cell type-specific expression programs with genes differentially expressed in the Pdx1 knockdown mutants, we were able to identify and refine likely gene targets specific to the *Sp-Pdx1/Sp-Brn1/2/4* neuronal cell type. In total, we found 249 genes belonging to the *Sp-Pdx1/Sp-Brn1/2/4* neuron subcluster (9) that were differentially expressed in the *Pdx1* knockdown dataset, with 65 % of the targets being downregulated (*Figure 7C*). Among the downregulated genes, we found key transcription factors involved in

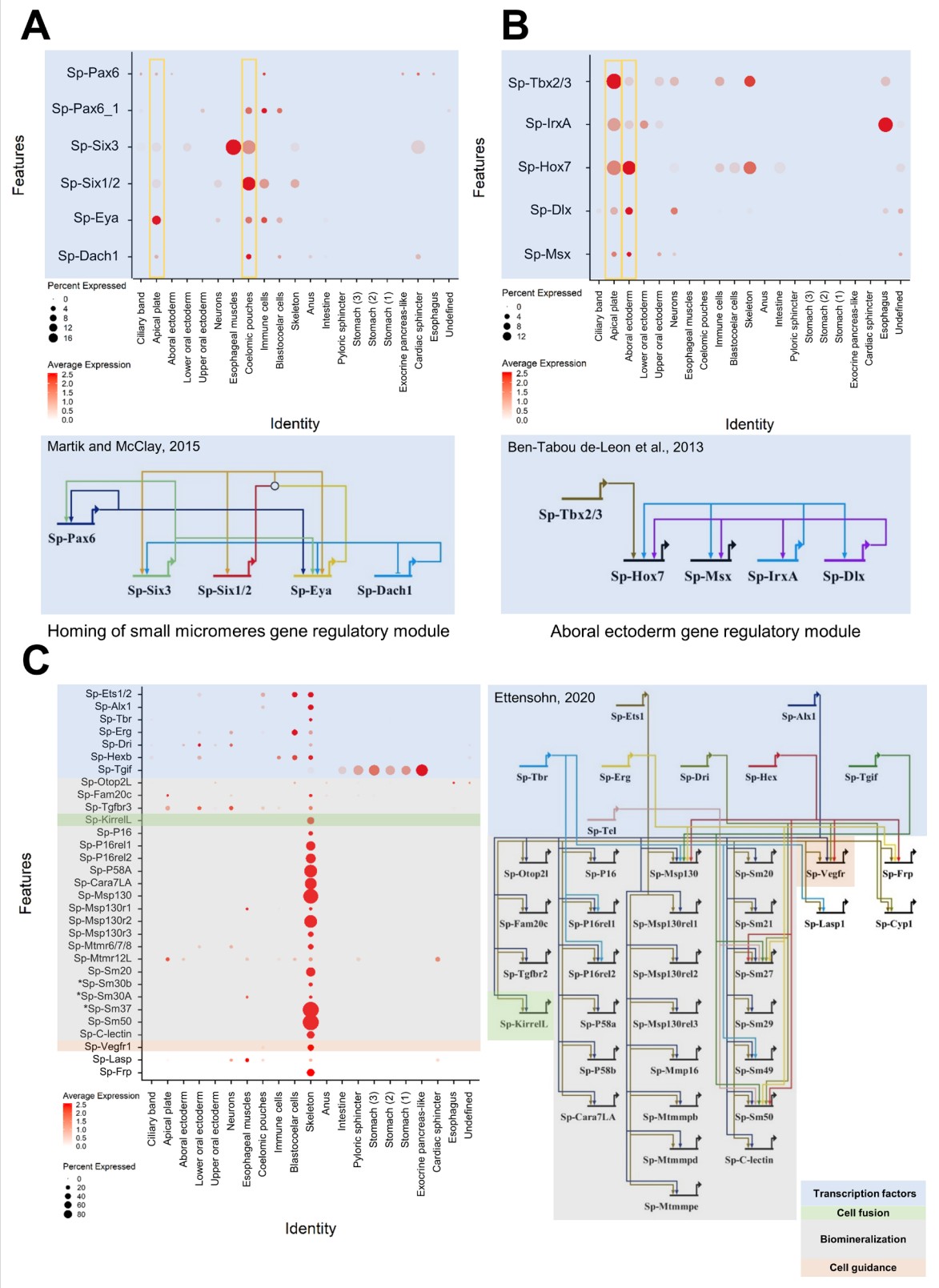

**Figure 5.** Validation of preexisting GRNs and putative novel function of specific gene regulatory modules. (**A**) Dotplot showing the mRNA localization of genes involved in the homing of small micromeres to the coelomic pouch and novel apical plate domain. (**B**) Dotplot of aboral ectoderm regulatory module genes showing novel apical plate expression. (**C**) Pre-gastrula gene regulatory network enriched in skeletal cells of the sea urchin pluteus larva. Asterisks indicate larval genes involved in biomineralization, putative members of this GRN.

*Figure 5 continued on next page*

*Figure 5 continued*

The online version of this article includes the following figure supplement(s) for figure 5:

**Figure supplement 1.** Reconstruction of the molecular signature of the gut at a single-cell resolution.

neuronal differentiation, including *Sp-Brn1/2/4* and *Sp-Otp*, as well as terminal differentiation genes important in neuronal signaling, such as *Sp-An, Sp-Ngfffap, Sp-Th,* and *Sp-Chat* (*Figure 7C*).

## Discussion

Cell type identity is determined by the differential use of genomic information among cells. Unraveling the distinct transcriptomic signatures of cell types yields valuable insight into their function, as well as their evolutionary and developmental origins. In recent years, single cell transcriptomics has emerged as a powerful and unbiased approach for characterizing cell type diversity across a wide variety of animal taxa, with studies spanning from insects (*Davie et al., 2018*; *Severo et al., 2018*; *Cho et al., 2020*), to cnidarians (*Sebé-Pedrós et al., 2018*; *Siebert et al., 2019*), planarians (*Swapna et al., 2018*; *Zeng et al., 2018*) and sea squirts (*Sharma et al., 2019*; *Cao et al., 2019*), as well as vertebrates such as zebrafish (*Wagner et al., 2018*; *Chestnut et al., 2020*), mice (*Nestorowa et al., 2016*; *Jung et al., 2019*; *Ximerakis et al., 2019*; *Yu et al., 2019*; *Qi et al., 2020*), and humans (*Yu et al., 2019*; *Qi et al., 2020*; *Esaulova et al., 2020*; *Zhao et al., 2020*).

The sea urchin embryo has served as a valuable model for understanding cell type molecular specification and differentiation via gene regulatory networks. Despite this, knowledge of later stages of development, including larval cell types, is limited. Here, we used single-cell RNA sequencing to generate a detailed atlas of cell types of the early pluteus larva, and to unravel the neuronal diversity at this critical stage that marks the end of embryogenesis and the beginning of the larval life cycle.

### Cellular diversity of the *S. purpuratus* pluteus larva

Conducting scRNA-seq on isolated *S. purpuratus* early pluteus cells we initially identified 21 genetically distinct cell clusters from 19.699 cells (*Figure 1A*), that consist of derivatives from all three germ layers. Similar ectodermal, mesodermal and endodermal representatives have also been reported at a single cell resolution in previous studies, carried out at either earlier developmental stages or different sea urchin species, highlighting the power of this technique to capture fine and dynamic developmental processes during sea urchin development (*Foster et al., 2020*; *Massri et al., 2021*). The cells comprising the 21 cell type families are expressing a total of 15,578 WHL genes (transcriptome models) and 12,924 genes (SPU gene models). Notably, the computationally identified number of cells per cluster did not correlate well with their actual distribution in the larva. This could be a bias linked to the dissociation process, as for instance skeletal structures due to their calcite and syncytial nature are the last to dissociate and hence cells can get trapped within the debris and thus be underestimated in our datasets. Nonetheless, the cell type families and cell types identified in our study include all known larval cell types, suggesting we obtained a sufficient number of cells to comprehensively survey cell diversity at this developmental stage. Further, the total number of genes expressed in our data was 15.579, in relatively close agreement to the 16.500 genes expressed at the end of *S. purpuratus* embryogenesis (*Tu et al., 2014*). Our results reveal a rich tapestry of cell type families within the early pluteus larva. In particular, our study reveals unprecedented transcriptional diversity among cells of the pluteus larva that relates to feeding and digestion, as seen by the large representation of such clusters in our dataset. This includes two distinct oral ectoderm cell clusters, as well as distinct cell expression programs for the esophagus, cardiac and pyloric sphincters, exocrine pancreas-like cells, three distinct stomach cell clusters, intestine and anus. We also identified the mesodermally derived esophageal muscle cell type that ensures the proper function of the digestive apparatus by regulating the flow of the water containing food within the different compartments of the gut.

Cell type family tree reconstruction reveals two well supported groups (*Figure 3—figure supplement 1*). One group represents the endoderm. It comprises a core of closely related cell type families (stomach cells 1,2,3, pyloric and cardiac sphincter, and exocrine pancreas-like cells) and the more distant subgroup of intestinal and anal cells. The coherence of this group is also apparent from the transcription factor heatmap (*Figure 4—figure supplement 2*), which reveals numerous factors shared between subsets or the entire endodermal group such as *Klf15* that is known to control lipid and

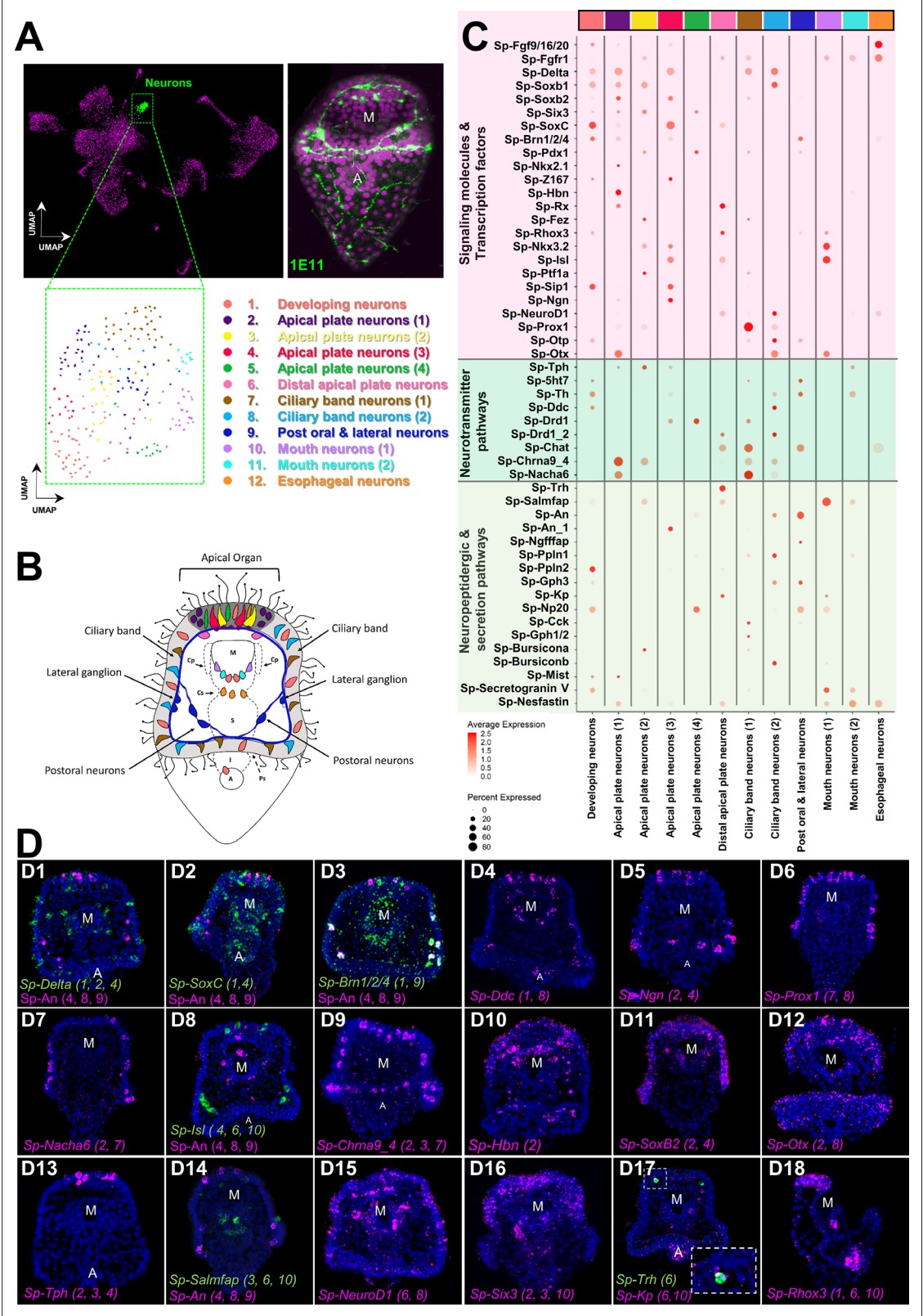

**Figure 6.** Neuronal complexity of the three dpf *S. purpuratus* larva. (**A**) (From left to right and top to bottom) UMAP highlighting the neurons cluster (green), immunohistochemical detection for the paneuronal sea urchin marker 1E11 (green), UMAP showing the 12 distinct neuronal subclusters. (**B**) Schematic representation of the three dpf pluteus larva showing the localization of neuronal subclusters (colors as in A). (**C**) Dotplot of signaling molecules, transcription factors, and neurotransmitters involved in sea urchin neuronal function and neurogenesis (colors as in A). (**D**) FISH of *S.*

*Figure 6 continued on next page*

*Figure 6 continued*

*purpuratus* three dpf larvae with antisense probes for the neuronal genes *Sp-Delta* (**D1**), *Sp-SoxC* (**D2**), *Sp-Brn1/2/4* (**D3**), *Sp-Ddc* (**D4**), *Sp-Ngn* (**D5**), *Sp-Prox1* (**D6**), *Sp-Nacha6* (**D7**), *Sp-Isl* (**D8**), *Sp-An* (**D8 and D14**), *Sp-Chrna9_4* (**D9**), *Sp-Hbn* (**D10**), *Sp-SoxB2* (**D11**), *Sp-Otx* (**D12**), *Sp-Tph* (**A13**), *Sp-Salmfap* (**D14**), *Sp-NeuroD1*(D15), *Sp-Six3* (**D16**), *Sp-Trh* (**D17**), *Sp-Kp* (**D17**), and *Sp-Rhox3* (**D18**). FISH shown in figures D1-3 are paired with immunohistochemical detection of the neuropeptide Sp-An. Nuclei are labeled with DAPI (in blue). All images are stacks of merged confocal Z sections. A, anus; M, mouth.

The online version of this article includes the following figure supplement(s) for figure 6:

**Figure supplement 1.** Subclustering of the skeletal cell type family.

**Figure supplement 2.** Subclustering of the immune cells cluster.

**Figure supplement 3.** Co-localization of *Sp-Fgf9/16/20* and *Sp-FgfR1* in the cardiac sphincter region.

glucose metabolism in the liver (*Oishi and Manabe, 2018*). The other group represents ectoderm. It contains apical plate, ciliary band, the aboral and upper oral ectoderm cells, and neurons. Again, this group manifests in shared transcription factor expression including *Klf2/4*, *Tbx2/3*, and *SoxB1* and *SoxB2*. Unexpectedly, this group firmly comprises some cell type families that are usually not considered ectodermal, such as the immune cells and the esophageal cells (*Figure 3—figure supplement 1*). Outside these two groups, the remaining cell type families do not assemble into a well-supported group. Instead, they exhibit molecular affinities to either the endodermal or ectodermal clades.

Beyond our initial cell clusters, we used subclustering to uncover diversity among neuronal, PMC, and immune cells in the pluteus larva. Among immune cells, the presence of two pigment cell subclusters is in line with findings by Perillo and colleagues that revealed two such populations (*Perillo et al., 2020*). However, in our study, we found that only one cluster is *Sp-Gcm* positive, in contrast to their findings which found both of their clusters expressed *Sp-Gcm*. This difference could be a result of the different approaches used to identify the different cell types or due to transient expression of *Sp-Gcm* in the additional pigment cell type.

The larval nervous system has been among the first echinoderm cell types to be characterized at a molecular level and yet the exact number of neuronal subtypes is still not clear. Extensive work has been done on identifying the molecular pathways guiding neuronal specification and the genes active during neuronal differentiation; however, this is limited to describing general neuronal categories. Most of the current information on the different neuronal types relies on detection of specific neuropeptides, neurotransmitters and enzymes involved in their biosynthesis. The most recent estimate of neuronal diversity used neuropeptidergic content to identify seven distinct neuronal types (*Wood et al., 2018*). Our study supports and refines this earlier work by providing a comprehensive and unbiased survey of neuronal diversity in the early pluteus larva of *S. purpuratus*.

Neurons in *S. purpuratus* arise from two ectodermal neurogenic regions (ciliary band and apical domain) and from the anteriormost part of the foregut (*Garner et al., 2016*; *McClay et al., 2018*; *Wei et al., 2011*), which is derived from endoderm (mouth neurons and esophageal neurons, see *Figure 6A*). Different neuronal types from these domains arise at different developmental time points, although by 3 dpf most larval neurons are thought to be present and patterning diverse larval domains. Notably, our initial clustering analysis recovered neurons as a single cluster, suggesting they share a common molecular signature regardless of their developmental origin. Subclustering of these revealed twelve distinct neuronal populations, which we were able to trace back on the larva. By doing so, we identified (i) one cluster consisting of neurons that apparently are undergoing differentiation as judged by the combinatorial expression of *Delta*, *Soxc*, *Brn1/2/4* and terminal differentiation gene markers, (ii) four associated with the apical domain, one of which is expressing the same combination of TFs mentioned above, suggesting it might also be undergoing differentiation, (iii) one matching the distal apical plate neurons, (iv) two corresponding to ciliary band neurons, (v) two located in the rim of the mouth, (vi) one associated with esophageal structures, and (vii) one corresponding to the post-oral and lateral neurons. Thus, our study greatly enhances our knowledge of neuronal diversity in the *S. purpuratus* larva, nearly doubling the number of known neuronal cell types at this developmental stage.

Lastly, we also found one cell cluster (undefined) that could not be traced back to a specific domain. Based on the numbers and distributions of the features and transcript numbers that are comparable to the ones of real cell type families, such as the esophageal cell type family, we speculate that this

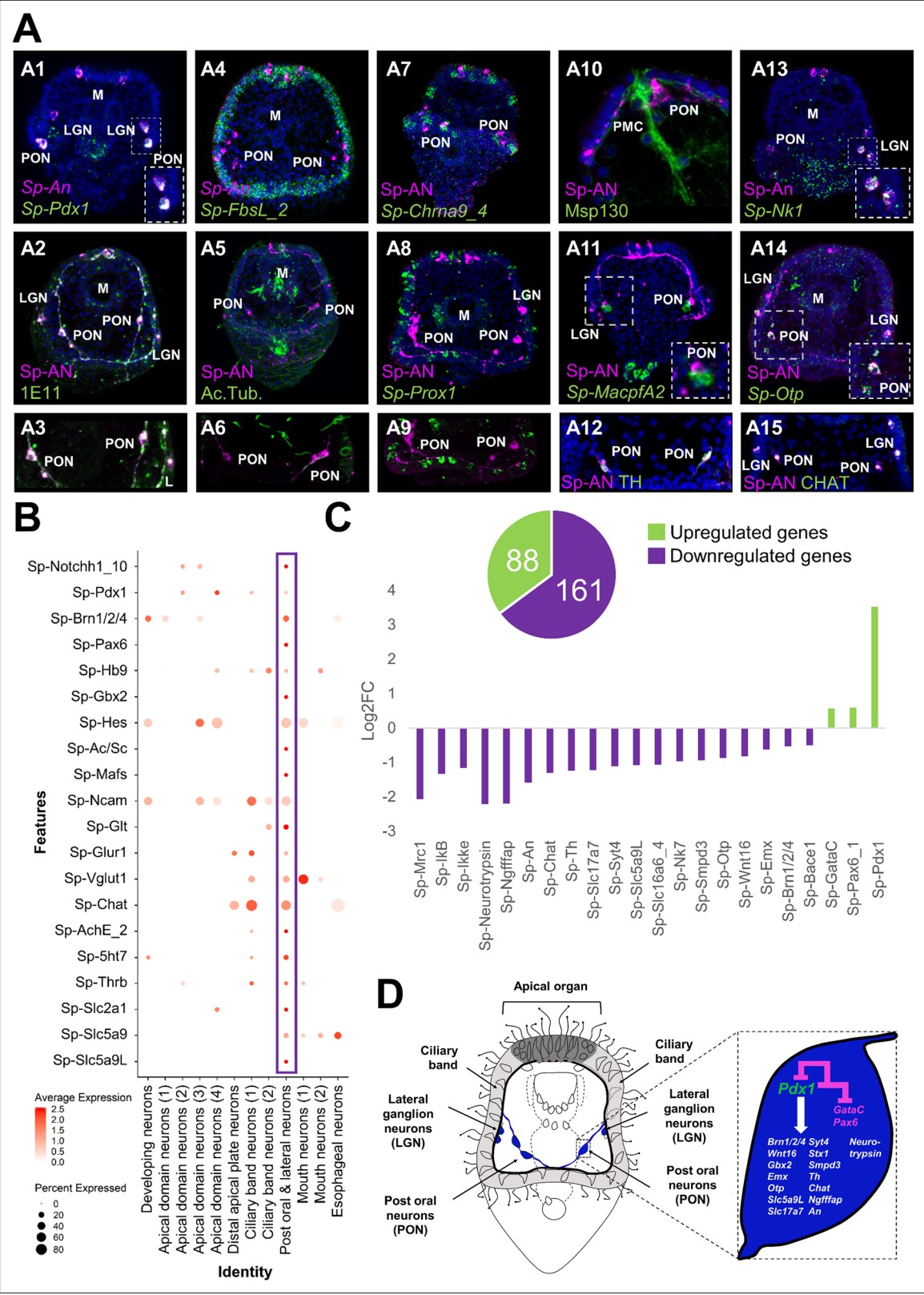

**Figure 7.** ScRNA-seq reveals a Pdx-1-dependent neuroendocrine cell type. (**A**) Molecular characterization of a *Sp-Pdx1*/*Sp-Brn1/2/4* double positive neuronal population. (**A1**) Double FISH of *S. purpuratus* 3 dpf larvae with specific antisense probes for *Sp-Pdx1* and *Sp-An*. (**A2**) Double immunohistochemical detection of the Sp-An and Synaptotagmin (1E11) proteins. (**A3**) Close up caption of the Sp-An PON neurons shown in A2. (**A4**) Double FISH of *S. purpuratus* three dpf larvae with specific antisense probes for *Sp-FbsL_2* and Sp-An. Double immunohistochemical detection of

*Figure 7 continued on next page*

Figure 7 continued

the Sp-An and acetylated tubulin proteins. (**A6**) Close-up caption of the Sp-An PON neurons shown in A5. (**A7**) FISH of *S. purpuratus* three dpf larvae with a specific antisense probe for *Sp-Chrna9_4* paired with immunodetection of Sp-An. (**A8**) FISH of *S. purpuratus* three dpf larvae with a specific antisense probe for *Sp-Prox1* paired with immunohistochemical detection of Sp-An. (**A9**) Close-up caption of the Sp-An PON neurons shown in A8. (**A10**) Double immunohistochemical staining for the neuropeptide Sp-An and the skeletal cells marker Msp130. (**A11**) FISH of *S. purpuratus* three dpf larvae with a specific antisense probe for *Sp-MacpfA2* paired with immunohistochemical detection of Sp-An. (**A12**) Double immunohistochemical staining for the neuropeptide Sp-An and the enzyme Sp-TH. (**A13**) Double FISH of *S. purpuratus* three dpf larvae with specific antisense probes for *Sp-Nk1* and *Sp-An*. (**A14**) FISH of *S. purpuratus* three dpf larvae with a specific antisense probe for *Sp-Otp* paired with immunohistochemical detection of Sp-An. Double immunohistochemical staining for the neuropeptide Sp-An and the enzyme Sp-Chat. Nuclei are labeled with DAPI (in blue). All images are stacks of merged confocal Z sections. LGN, lateral ganglion neurons; M, Mouth; PON, Post-oral neurons. (**B**) Dotplot of genes important in endocrine pancreas differentiation and function in vertebrates. (**C**) Bar plot of selected Sp-Pdx1 target genes in the *Sp-Pdx1/Sp-Brn1/2/4* positive population as revealed by differential RNA sequencing analysis of *Sp-Pdx1* knockdown larvae. (**D**) Schematic representation of the Sp-Pdx1/Sp-Brn1/2/4 neurons genetic wiring as revealed by the combination of scRNA-seq and differential RNA-seq analysis after *Sp-Pdx1* knockdown, highlighting an important role of *Sp-Pdx1* in the differentiation of this cell type.

The online version of this article includes the following figure supplement(s) for figure 7:

**Figure supplement 1.** Pancreatic genes expression significance testing in the 'post oral and lateral neurons' subcluster.

represents a real cell type family and not empty droplets. (*Figure 1—figure supplement 1E*). Overall, this cluster exhibited greatest transcriptional similarity to cell types derived from the ectoderm (*Figure 1—figure supplement 1F*), suggesting it may be of ectodermal origin. Consistent with this, a recent study revealed a similar uncharacterized ectodermal cell type (*Perillo et al., 2020*), further suggesting this cell population exists and is not solely an artifact of our analysis. Taking into account the great plasticity and regeneration capability of the sea urchin larva, it is also possible that this non-differentiated ectodermal cell type is a progenitor population in stasis, waiting to be activated. Future studies are necessary to validate its identity, function, and origin during larval development.

## Sp-Pdx1 as a regulator of neuroendocrine fate

Morphogenesis and organogenesis rely on the hierarchical control of gene expression as encompassed in the GRNs. Although sea urchins diverged from chordates prior to the origin of the pancreas, we have previously demonstrated that the neurogenic and neuronal territories of the sea urchin embryo and larva have a strong pancreatic-like molecular signature (*Perillo et al., 2018*). Interestingly, one neuronal population was found to express *Sp-Pdx1* and *Sp-Brn1/2/4*, as well as the echinoderm-specific neuropeptide An. *Pdx1* in mammals is essential for proper pancreatic formation, β-cell differentiation, and regulation of the mature β-cells physiology and function (*Hui and Perfetti, 2002*). On the other hand, the vertebrate orthologue of *Sp-Brn1/2/4*, *Brn4* is expressed in the α pancreatic cells, where it acts as a key differentiation factor of this lineage (*Hussain et al., 2002*).

The gene regulatory cascade leading to the endocrine pancreas formation in vertebrates has been described in great detail and several transcription factors (Hes, Ascl1, Pax6, Hb9, Mafs, Gbx2), and signaling molecules (Notch) have been characterized as essential for this process (*Ishibashi et al., 1995*; *de la Pompa et al., 1997*; *Li et al., 1999*; *Iso et al., 2003*; *Mizusawa et al., 2004*; *Zaret and Grompe, 2008*; *Arkhipova et al., 2012*; *Hart et al., 2013*; *Tritschler et al., 2017*; *Mitchell et al., 2017*; *Buckle et al., 2018*; *de la Pompa et al., 1997*; *Iso et al., 2003*).

Here, we dissected the molecular fingerprint of the *Sp-Pdx1/Sp-Brn1/2/4* neuronal type and identified the presence of genes involved in pancreatic development as well as of genes known to be expressed in both endocrine pancreatic cells and neurons. From our scRNA-seq analysis, it is evident that *Sp-Notch*, *Sp-Pax6*, *Sp-Hb9*, *Sp-Hes*, *Sp-Ac/Sc* (the orthologue of ASCL1), *Sp-Mafs* and the recently re-annotated *Sp-Gbx2* (previously annotated as *Sp-Nk7*- https://new.echinobase.org) are all expressed in the *Sp-Pdx1/Sp-Brn1/2/4* neurons. This suggests that these neurons of a non-chordate deuterostome have a gene regulatory machinery similar to the endocrine pancreas cells.

It has also been demonstrated that both endocrine pancreas and neuronal cells share similar features and are able to produce and respond to several neuronal genes and neurotransmitters. Interestingly, we were able to identify these shared components in our *Sp-Pdx1/Sp-Brn1/2/4* neurons. For instance, the ortholog of the neural cell adhesion molecule *Ncam* known to be produced in the nervous system and endocrine cells of the rat is also expressed in this cluster (*Langley et al., 1987*). Moreover, these neurons also express genes encoding members of the glutamate signaling pathway

(*Sp-Glt*: glutamate synthase; *Sp-Vglut1*: glutamate transporter; *Sp-Glur1*: glutamate receptor), which in mammals are involved in glucose-responsive insulin secretion (*Gonoi et al., 1994*; *Maechler and Wollheim, 1999*), and tyrosine hydroxylase (*Th*), the rate-limiting enzyme of catecholamine biosynthesis, which is present in the endocrine pancreas of multiple species (*Teitelman et al., 1993*; *Iturriza and Thibault, 1993*). Furthermore, *Sp-Pdx1/Sp-Brn1/2/4* neurons express choline acetyltransferase (*Chat*), which has been found to be highly expressed in human pancreatic islets and may be essential for the stimulation of insulin secretion by the neighboring β-cells (*Rodriguez-Diaz et al., 2011*).

Additionally, serotonergic signaling is believed to be involved in the regulation of insulin secretion as several serotonin receptors have been found to be expressed in human pancreatic islets (*Amisten et al., 2015*). Transcripts of the sea urchin serotonin receptor *Sp-5ht7* are present in *Sp-Pdx1/Sp-Brn1/2/4* neuronal population suggesting that serotonin might regulate the neurotransmitter/neuropeptide secretion in a similar way as for insulin. Similarly, transcripts of the Thyroid hormone receptor B are also found in this population, suggesting that thyroid hormone signaling may play a role to its differentiation similar to the one in murine endocrine pancreas differentiation (*Aiello et al., 2014*). The hormone secretion mediated by pancreatic endocrine cells depends on their ability to detect changes in extracellular glucose levels. To this end, they are equipped with Glucose transporters and co-transporters (*Navale and Paranjape, 2016*; *Berger and Zdzieblo, 2020*). Our analysis revealed that the *Sp-Pdx1/Sp-Brn1/2/4* neurons produce transcripts of three glucose co-transporter genes (*Sp-Slc2a1*, *Sp-Slc5a9* and *Sp-Slc5a9L*) proposing that they are able to detect such changes in glucose levels similarly to the endocrine pancreas cells.

## Evolutionary considerations

Taken together, our data show that the sea urchin *Sp-Pdx1/Sp-Brn1/2/4* neurons uniquely express a combination of key genes that are necessary for the endocrine pancreas differentiation and function in the vertebrates (*Figure 7*, *Figure 7—figure supplement 1*). These shared molecular features of *Sp-Pdx1/Sp-Brn1/2/4* neurons and pancreatic cells suggest these represent features of a cell type present in the deuterostome ancestor. What was the nature of these cells and what happened in the evolution of the descendant lineages so that the molecular characteristics of these cells are found in extant cell types as divergent as ectodermal neurons and pancreatic cells?

One possibility is that the ancestral deuterostomes already possessed a *Sp-Pdx1/Sp-Brn1/2/4* neuronal cell type similar to the one found in today's sea urchin, and that the regulatory program of this cell type was then co-opted into the endodermal lineage in the vertebrates. The co-option and incorporation of gene regulatory elements to different developmental or morphogenetic context has been postulated before and has the potential to give rise to diverse cell types (*Monteiro, 2012*; *Preger-Ben Noon and Frankel, 2015*; *Martik and McClay, 2015*; *Hu et al., 2018*; *Morgulis et al., 2019*; *McQueen and Rebeiz, 2020*; *Cary et al., 2020*). In particular, it has been hypothesized that β pancreatic cells arose during evolution by co-option of a preexisting neuronal cell type program into the pancreas developmental lineage ( *Arntfield and van der Kooy, 2011*) based on the many physiological, morphological and molecular features endocrine pancreatic cells share with neurons (*Alpert et al., 1988*, *Eberhard et al., 2013*). However, given the long list of genes shared between neurons and pancreatic cells, this would have necessarily represented a massive co-option event profoundly altering the identity and function of the receiving cells. Notably, a flexibility toward the co-option of a neuroendocrine phenotype has also been reported in the vertebrate lineage in the case of the adrenal medulla derived PC12 cell line, that upon stimulation with NGF, adopts a neurosecretory fate (*Fujita et al., 1989*).

A second possibility is that both ectodermal neurons and endodermal pancreatic cells are direct evolutionary descendants of the same precursor cell, representing sister cell types (*Arendt, 2008*). This explanation would avoid co-option; instead, it postulates a direct evolutionary link between cell types derived from different germ layers - ectoderm versus endoderm. Surprising at first, a similar link between digestive and neuronal-type expression profiles has been reported repeatedly. For example, it appears to hold true for choanocytes in the early-branching sponges (*Musser et al., 2021*), indicating that the family of sponge choanocytes might be related to the hypothesized digestive-neuronal precursors. In the sea anemone *Nematostella vectensis*, pharyngeal cells give rise to both digestive cells and neurons (*Steinmetz et al., 2017*), and neuronal cell types and secretory gland cells exhibit related molecular profiles (*Sebé-Pedrós et al., 2018*). In fact, based on these findings a close

evolutionary link between endoderm and ectoderm has been postulated (*Steinmetz et al., 2017*). Finally, motor neurons in the vertebrate ventral neural tube and pancreatic islet cells share a specific combination of transcription factors including *pax6, nkx6,* and *hb9* (reviewed in *Arendt, 2021*). Overall, this second possibility is also in line with the hypothesis that at least subsets of neuron types in animal nervous systems may have originated from a mucociliary sole that initially harbored a neuro-secretory network of cells with digestive and communicative functions (*Arendt et al., 2015*; *Arendt, 2021*). Via division of labor, these precursor cell types may have then given rise to both digestive as well as neuronal cell types in different animal phyla.

## Materials and methods
### Animal husbandry and culture of embryos
Adult *Strongylocentrotus purpuratus* individuals were obtained from Patrick Leahy (Kerckhoff Marine Laboratory, California Institute of Technology, Pasadena, CA, USA) and maintained in circulating seawater aquaria at Stazione Zoologica Anton Dohrn in Naples. Gametes were obtained by vigorous shaking of the animals. Embryos and larvae were cultured at 15 °C in filtered Mediterranean Sea water diluted 9:1 with deionized water.

### Larvae dissociation
Dissociation of the three dpf *Strongylocentrotus purpuratus* plutei into single cells was performed according to an adaptation of several protocols (*McClay, 1986*; *McClay, 2004*; *Juliano et al., 2014*). Larvae were collected, concentrated using a 40 µm Nitex mesh filter and spun down at 500 g for 5 min. Sea water was removed and larvae were resuspended in $Ca^{2+}$ $Mg^{2+}$-free artificial sea water. Larvae were spun down at 500 g for 5 min and resuspended in dissociation buffer containing 1 M glycine and 0.02 M EDTA in $Ca^{2+}$ $Mg^{2+}$-free artificial sea water. Larvae were incubated for 10 min on ice and mixed gently via pipette aspiration every 2 min. From that point and onwards the progress of dissociation was monitored. Dissociated cells were spun down at 700 g for 5 min and washed several times with $Ca^{2+}$ $Mg^{2+}$-free artificial sea water. Cell viability was assessed via using Propidium Iodide and Fluorescein diacetate and only specimens with cell viability ≥ 90 % were further processed. Single cells were counted using a hemocytometer and diluted according to the manufacturer's protocol (10 x Genomics). Throughout this procedure samples were kept at 4 °C.

### Single-cell RNA sequencing and data analysis
Single cell RNA sequencing was performed using the 10 x Genomics single-cell capturing system. Specimens from four independent biological replicates, ranging from 6000 to 20,000 cells, were loaded on the 10 X Genomics Chromium Controller. Single cell cDNA libraries were prepared using the Chromium Single Cell 3' Reagent Kit (Chemistries v2 and v3). Libraries were sequenced by GeneCore (EMBL, Heidelberg, Germany) for 75 bp paired-end reads (Illumina NextSeq 500), resulting in a mean of 88 M reads. Cell Ranger Software Suite 3.0.2 (10x Genomics) was used for the alignment of the single-cell RNA-seq output reads and generation of feature, barcode and matrices. The genomic index was made in Cell Ranger using the *S. purpuratus* genome version 3.1 (*Sea Urchin Genome Sequencing Consortium et al., 2006*; *Kudtarkar and Cameron, 2017*). Cell Ranger output matrices for four biological and two technical replicates were used for further analysis in Seurat v3.0.2 R package (*Stuart et al., 2019*). The analysis was performed according to the Seurat scRNA-seq R package documentation (*Butler et al., 2018*; *Stuart et al., 2019*). Genes that are transcribed in less than three cells and cells that have less than a minimum of 200 transcribed genes were excluded from the analysis. The cutoff number of transcribed genes was determined based on feature scatter plots and varies depending on the replicate. Of the 29,130 cells estimated by Cell Ranger, 19,699 cells passed the quality checks and were further analyzed. Datasets were normalized and variable genes were found using the vst method with a maximum of 2000 variable features. Data integration was performed via identification of anchors between the six different objects. Next the datasets were scaled and principal component (PCA) analysis was performed. Nearest Neighbor (SNN) graph was computed with 20 dimensions (resolution 1.0) to identify the clusters. Uniform Manifold Approximate and Projection (UMAP) was used to perform clustering dimensionality reduction. Cluster markers were found using the genes that are detected in at least 0.01 fraction of min.pct cells in the two clusters.

Transcripts of all genes per cell type were identified by converting a Seurat DotPlot with all these transcripts as features into a table (ggplot2 3.2.0 R package). Subclustering analysis was performed by selecting a cell type family of interest and performing similar analysis as described above. All resulting tables containing the genes transcribed within different cell type families were further annotated adding PFAM terms (*Trapnell et al., 2010*; *Finn et al., 2014*) for associated proteins, gene ontology terms and descriptions from Echinobase (*Kudtarkar and Cameron, 2017*).

Cell type tree reconstruction was performed in R (version 4.0.3) using the neighbor-joining tree function in package 'ape' (version 5.4–1). Two final trees were constructed, one that derived pairwise expression distances between clusters using all expressed genes, and a second in the analysis was restricted to only expressed transcription factors. A gene was defined as 'expressed' when it had an average expression of greater than three transcripts per million in at least one of the 21 major clusters. For each tree, pairwise distances were calculated as 1 – spearman correlation between each of the 20 major clusters, excluding the undefined cells, using the spearman.dist function in the bioDist package (version 1.60.0). To assess the support of each bifurcation in the tree, we performed 10,000 bootstrap replicates using the boot.phylo function in 'ape'. Although not reported in this manuscript, we also tested the effect of using different distance metrics and normalization methods on tree topology. Specifically, we reconstructed trees using distances derived from Pearson correlations of either log or square root normalized average expression values, and Euclidean distances of scaled log-normalized expression values. In general, similar topologies were found in all analyses, recovering clades of endodermal and ectodermal derivatives. Mesodermal derivatives often, but not always, grouped together, and the position of the neuron cluster moved in and out of the clade of ectodermal derivatives. This latter result may possibly arise because of conflicting signals resulting from cell type heterogeneity (i.e. different neuron types) within the cluster.

In order to assess the significance of identifying the homologs of the vertebrate genes involved in the development of the pancreas in the 'post oral and lateral neurons' sub-population of neuronal cells, the normalized counts from all the transcripts present in the neuronal subcluster were extracted in the form of a matrix using GetAssayData command from Seurat v3.0.2. The counts within this matrix were then randomized with 100,000 iterations using the randomizeMatrix function from the picante v1.8.2 package (*Kembel et al., 2010*) with the independent swap null model. A new Seurat object was created, which is the same as the neuronal subclustering object, however the actual transcript normalized counts were replaced with the randomized ones. Then the average expression of the 20 pancreatic genes was extracted from both from actual neuronal subclustering Seurat object and the one with randomized counts using AverageExpression function to visualize the expression of these pancreatic genes in the 'post oral and lateral neurons' subcluster in actual data and randomized data. This process was looped 500 times for randomized data to then calculate the mean number of pancreatic genes expressed in the 'post oral and lateral neurons' subcluster to obtain an estimate of the expected number of genes expressed, rounded to the closest integer, if their expression was found in this subcluster by random. The number of genes expressed in the actual data and the randomized data was then used to generate proportions of the number of expressed and not expressed pancreatic genes in this neuronal subcluster. Then, we have performed a chi-squared test within R (*Hope, 1968*) to show that the proportion from the actual data proportion is significantly different to the randomized data proportion with p-value less than 0.002.

## Whole mount RNA fluorescent in situ hybridization

Fluorescent in situ hybridization was performed as outlined in *Perillo et al., 2021*. Fluorescent signal was developed via using fluorophore conjugated tyramide technology (Perkin Elmer, Cat. #NEL752001KT). Antisense probes were transcribed from linearized DNA and labeled either during transcription via using digoxigenin-11-UTP nucleotides, or post-transcriptionally by using Fluorescein (Mirus Bio, Cat. #MIR3200) or DNP (Mirus Bio, Cat. #MIR3825) following the manufacturer's instructions. Probes for *Sp-Pdx1, Sp-Cdx, Sp-ManrC1A, Sp-Six1/2, Sp-Fgf9/16/20, Sp-Brn1/2/4, Sp-Ngn, Sp-Isl, Sp-NeuroD1, Sp-Pks1, Sp-Soxb2, Sp-An, Sp-Trh, and Sp-Salmfap* were produced as previously published [*Sp-Pdx1, Sp-Cdx* (*Cole et al., 2009*), *Sp-ManrC1A* (*Annunziata et al., 2014*), *Sp-Six1/2, Sp-Fgf9/16/20* (*Andrikou et al., 2015*), *Sp-Brn1/2/4* (*Cole and Arnone, 2009*), *Sp-Ngn, Sp-Isl, Sp-NeuroD1* (*Perillo et al., 2018*), *Sp-Pks1* (*Perillo et al., 2020*), *Sp-SoxB2* (*Anishchenko et al., 2018*), *Sp-An, Sp-Trh, Sp-Salmfap* (*Wood et al., 2018*)]. Primer sequences used for cDNA isolation

and probes synthesis are in *Supplementary file 2*. Specimens were imaged using a Zeiss LSM 700 confocal microscope.

## Immunohistochemistry (IHC)

Immunohistochemical staining or IHC paired with FISH was performed as described in *Perillo et al., 2021*. Briefly three dpf plutei were fixed in 4 % paraformaldehyde (PFA) in filtered sea water (FSW) for 15 min at room temperature (RT). FSW was removed and samples were incubated in 100 % methanol for 1 min at RT, washed multiple times with phosphate buffer saline with 0.1 % Tween 20 (PBST) and incubated blocking solution containing 1 mg/ml Bovine Serum Albumin (BSA) and 4 % sheep in PBST for 1 hr. Primary antibodies were added in the appropriate dilution and incubated for 1 hr and 30 min at 37 °C. Anti-acetylated alpha tubulin (Sigma-Aldrich T67930) was used to label cilia and microtubules (1:200), Anti-Msp130 (gift from Dr. David R. McClay) to label skeletogenic cells (undiluted), 1E11 (gift from Dr. Robert Burke) to mark the nervous system (1:20), 5c7 (gift from Dr. David R. McClay) to label the endoderm (undiluted), Sp-An to label the post-oral and lateral neurons (1:250), Sp-Th (Sigma-Aldrich AB152) to label catecholaminergic neurons (1:100) and Sp-Chat (GeneTex GXGTX113164S) to label cholinergic neurons (1:100). Specimens were washed multiple times with PBST and incubated for 1 hr with the appropriate secondary antibody (AlexaFluor) diluted 1:1000 in PBST. Larvae were washed several times with PBST and imaged using a Zeiss LSM 700 confocal microscope.

## EdU labeling paired with immunohistochemistry

In order to understand the spatial distribution of proliferating cells across the putative broad cell types, cell proliferation assays were carried out using Click-It EdU Cell Proliferation Kit for Imaging Alexa Flour 647 (Thermo Fisher Scientific). Larvae were treated with EdU at a final concentration of 10 μM in FSW and let to grow for 2 hr. Samples were fixed in 4 % PFA in FSW for 15 min (RT) and washed several times with PBST. PBST was removed, replaced by 100 % Methanol for 1 min (RT) and followed by several washes with PBST. After this step, one can continue with either developing the EdU signal or performing immunohistochemistry as described above. In order to develop the EdU signal, the Click-iT reaction mix was prepared according to the manufacturer's guidelines. PBST was removed and the reaction mix was added to the samples for 30 min (RT). Larvae were washed several times with PBST, mounted and imaged using a Zeiss LSM 700 confocal microscope.

## Gene regulatory network draft

Gene regulatory modules and networks were drafted using the interactive tool for building and visualizing GRNs BioTapestry (*Longabaugh, 2012*).

## Acknowledgements

The authors thank Prof. Paola Oliveri (UCL) for providing critical input on the analysis and structure of the manuscript. We are grateful to Drs. David McClay and Robert Burke for kindly providing antibodies and Dr. Francesco Lamanna (ZMBH) for helping with the computational analysis. We also thank Davide Caramiello for taking care of the adult sea urchins, Dr. Giovanna Benvenuto (SZN) for microscopy assistance and the Arnone lab members Maria Cocurullo and Inés Fournon Berodia for their help in gene cloning and preparation of several RNA probes. We also thank Drs. Vladimir Benes and Bianka Baying (GeneCore, Heidelberg, Germany) for sequencing of our single cell datasets. Also we are grateful to the animal technician Emily Savage (EMBL, Heidelberg, Germany) for her precious assistance.

## Additional information

### Funding

| Funder | Grant reference number | Author |
|---|---|---|
| H2020 Marie Skłodowska-Curie Actions | 766053 | Periklis Paganos<br>Detlev Arendt<br>Maria Ina Arnone |
| H2020 European Research Council | 788921 | Jacob M Musser<br>Detlev Arendt |

The funders had no role in study design, data collection and interpretation, or the decision to submit the work for publication.

### Author contributions

Periklis Paganos, Conceptualization, Data curation, Formal analysis, Investigation, Methodology, Validation, Visualization, Writing - original draft, Writing - review and editing; Danila Voronov, Data curation, Formal analysis, Methodology, Writing - review and editing; Jacob M Musser, Data curation, Formal analysis, Funding acquisition, Methodology, Writing - review and editing; Detlev Arendt, Funding acquisition, Writing - review and editing; Maria Ina Arnone, Conceptualization, Funding acquisition, Investigation, Methodology, Project administration, Resources, Supervision, Visualization, Writing - review and editing

### Author ORCIDs

Periklis Paganos (iD) http://orcid.org/0000-0001-9525-4625
Danila Voronov (iD) http://orcid.org/0000-0002-2972-6484
Detlev Arendt (iD) http://orcid.org/0000-0001-7833-050X
Maria Ina Arnone (iD) http://orcid.org/0000-0002-9012-7624

### Decision letter and Author response

Decision letter https://doi.org/10.7554/eLife.70416.sa1
Author response https://doi.org/10.7554/eLife.70416.sa2

## Additional files

### Supplementary files

• Transparent reporting form

• Supplementary file 1. Differentially expressed genes per cell type family and putative target genes in the PDX1 positive neurons.

• Supplementary file 2. Primers used to generate specific antisense RNA probes.

### Data availability

Sequencing data (mapped reads) have been deposited in Dyrad under the unique identifier https://doi.org/10.5061/dryad.n5tb2rbvz.

The following dataset was generated:

| Author(s) | Year | Dataset title | Dataset URL | Database and Identifier |
|---|---|---|---|---|
| Paganos P, Voronov D, Musser J, Arendt D, Arnone MI | 2021 | Data from: Sp 3 dpf scRNA-Seq | http://dx.doi.org/10.5061/dryad.n5tb2rbvz | Dryad Digital Repository, 10.5061/dryad.n5tb2rbvz |

*Continued on next page*

The following previously published datasets were used:

| Author(s) | Year | Dataset title | Dataset URL | Database and Identifier |
|---|---|---|---|---|
| Lowe EK | 2016 | Data from: paraHox_analysis | https://github.com/elijahlowe/paraHox_analysis | Github, paraHox_analysis |
| Lowe EK | 2015 | S. purpurarus reads/ Sp-Lox 48 and 72 | https://osf.io/cbsxr/files/ | Open Science Framework, cbsxr |

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
