## [Editor Report]

This work provides a comprehensive analysis of cell state specification of a whole deuterostome organism, the sea urchin *Strongylocentrotus purpuratus*. It is also vigorous example for the use of single-cell sequencing to identify cell type homologies across evolution. The paper is thus of significant interest to scientists within the broad fields of developmental biology and evolution, as well as to the more specific communities of researchers that use the sea urchin as a model system or those interested in employing the single-cell mRNA-sequencing technology for "non-conventional" (and marine) molecular model systems.

---

## [Decision Letter]

**Decision letter after peer review:**

Thank you for submitting your article "Single cell RNA sequencing of the Strongylocentrotus purpuratus larva reveals the blueprint of major cell types and nervous system of a non-chordate deuterostome" for consideration by *eLife*. Your article has been reviewed by 3 peer reviewers, and the evaluation has been overseen by a Reviewing Editor and Marianne Bronner as the Senior Editor. The following individuals involved in review of your submission have agreed to reveal their identity: Pedro Martinez Serra (Reviewer #1); Roger Revilla-i-Domingo (Reviewer #2); Veronica Hinman (Reviewer #3).

Essential revisions:

As a short general note, please include line numbers in the revised version. It makes it easier for reviewers to refer to specific sentences they want to comment on.

1. In order to further emphasize the strength of the new data provided by this paper in characterizing the neuronal population that shows a "pancreatic" profile, the addition of a statistical analysis to the identification of 20 genes involved in the formation and function of vertebrate endocrine pancreas would add a major improvement. In other words, how many "pancreatic" genes could be expected to be found by chance within the transcriptional profile of this population?

2. Please add some comments on the conservation (or lack of conservation) of the connections shown in Figure 7D. If none of the connections shown in Figure 7D are known for the vertebrate pancreas, would this have any implications for the evolutionary discussion? Or could the connections discovered in the sea urchin be taken as potentially conserved connections that should be tested in the vertebrate pancreas in the future?

Alternatively, since the suggested interactions have not been proven (in this paper) consider to substitute this diagram for one (like in Figure 6B) in which only the neuroendocrine cells appear.

3. There are several new findings – that soxc, hbn are expressed in larval skeleton and *Fgf9*/16/20 in gut tissue that have not been reported previously, and contradict previous work. The authors should show clear double FISH images of these spatial expression patterns as it can be very easy to be misled in images to think that expression in the overlying ectoderm is actually internal. The images for soxc and FGF are not convincing. This is also especially important for *Fgf9*/16/20 as the authors argue this identifies and new neural cluster – also a new finding- and needs to be well supported with in-situ data.

In general, an improvement of the images from the double in situ will help to make the manuscript more convincing.

4. Are the 15578 "genes" describe actual gene models or transcripts, ie. are they distinct gene models or alternative transcripts.

5. Can the authors provide any additional evidence to support the classification of the unidentified cluster being an undifferentiated cell type. This has important significance for studies of regeneration and cell plasticity and will be important for future users of this dataset. Non-specific clusters can be driven by many clustering artifacts and e.g. metabolic and cellular processes, cell cycle.

---

## [Author Response]

Essential revisions:As a short general note, please include line numbers in the revised version. It makes it easier for reviewers to refer to specific sentences they want to comment on.1. In order to further emphasize the strength of the new data provided by this paper in characterizing the neuronal population that shows a "pancreatic" profile, the addition of a statistical analysis to the identification of 20 genes involved in the formation and function of vertebrate endocrine pancreas would add a major improvement. In other words, how many "pancreatic" genes could be expected to be found by chance within the transcriptional profile of this population?

We thank the reviewers for this suggestion. Statistical analysis has been performed showing that these 20 genes found in the *Pdx1/Brn1/2/4* neurons cannot be co-localized in this neuronal type by chance. The results of this analysis can be found in a new supplementary figure (Figure 7—figure supplement 1), while the way it was performed can be found in Materials and methods, section Single cell RNA-sequencing and data analysis and is as follows:

“In order to assess the significance of identifying the homologs of the vertebrate genes involved in the development of the pancreas in the "post oral and lateral neurons" sub-population of neuronal cells, the normalized counts from all the transcripts present in the neuronal subcluster were extracted in the form of a matrix using GetAssayData command from Seurat v3.0.2. […] Then, we have performed a chi-squared test within R (Hope, 1968) to show that the proportion from the actual data proportion is significantly different to the randomized data proportion with p-value less than 0.002.”

2. Please add some comments on the conservation (or lack of conservation) of the connections shown in Figure 7D. If none of the connections shown in Figure 7D are known for the vertebrate pancreas, would this have any implications for the evolutionary discussion? Or could the connections discovered in the sea urchin be taken as potentially conserved connections that should be tested in the vertebrate pancreas in the future?Alternatively, since the suggested interactions have not been proven (in this paper) consider to substitute this diagram for one (like in Figure 6B) in which only the neuroendocrine cells appear.

We thank the reviewers for their comment and agree with their point of view. Since these interactions have not yet been proven apart from an deRNA-seq level we choose to follow the reviewers’ recommendation and replace the GRN diagram with one depicting an example of the genes affected by Pdx1 MASO in the Pdx1/Brn124 double positive neuronal type. Furthermore, we use this schematic representation to highlight the potential role of *Sp-Pdx1* as an activator of neuronal fate and a crucial step for their differentiation process.

3. There are several new findings – that soxc, hbn are expressed in larval skeleton and Fgf9/16/20 in gut tissue that have not been reported previously, and contradict previous work. The authors should show clear double FISH images of these spatial expression patterns as it can be very easy to be misled in images to think that expression in the overlying ectoderm is actually internal. The images for soxc and FGF are not convincing. This is also especially important for Fgf9/16/20 as the authors argue this identifies and new neural cluster – also a new finding- and needs to be well supported with in-situ data.In general, an improvement of the images from the double in situ will help to make the manuscript more convincing.

We agree with the reviewers and have updated the figure by introducing new panels showing double FISH as well as FISH combined with immunostainings to better characterize the new expression patterns (Figure 2 panel D and Figure 6—figure supplement 3). In detail we added:

– Double FISH for *Sp-SoxC* and *Sp-Fgf9*/16/20 have been added to show the specific expression of SoxC in PMCs (Figure 2 D1-D4).

– A new improved FISH for *Sp-Hbn* paired with immunohistochemical detection for Msp130 has been added to show the expression of *Sp-Hbn* in the vertex PMCs (Figure 2 D5-D9).

– FISH for *Sp-Fgf9*/16/20 paired with immunohistochemical detection of the midgut and posterior gut marker Endo1 to clearly show the novel gut domains in which *Fgf9*/16/20 is expressed (Figure 2 D10-D13).

– Double FISH of *Sp-Fgf9*/16/20 and *Sp-Fgfr1* to show the possible location of the esophageal/cardiac sphincter neuronal type in which these two genes are predicted to be co-expressed (Figure 6—figure supplement 3).

4. Are the 15578 "genes" describe actual gene models or transcripts, ie. are they distinct gene models or alternative transcripts.

The number 15,578 corresponds to number of unique WHL (transcriptome models) corresponds to transcripts. In an attempt to better investigate the approximate number of genes expressed the number of unique SPU IDs (SPU gene models) corresponding to WHLs was analyzed and doing so we found 12,924 unique distinct or combination of SPU IDs corresponding to one gene. Moreover, we report that 1921 WHL IDs found in our data do not correspond to a SPU ID. We’ve updated this part of the manuscript, indicating each time the numbers of both WHL and SPU gene numbers.

5. Can the authors provide any additional evidence to support the classification of the unidentified cluster being an undifferentiated cell type. This has important significance for studies of regeneration and cell plasticity and will be important for future users of this dataset. Non-specific clusters can be driven by many clustering artifacts and e.g. metabolic and cellular processes, cell cycle.

We thank the reviewers for this comment. Throughout the manuscript we’ve presented several theories in the results and Discussion sections of what this cluster could correspond to suggesting also non-differentiated cells. However, we currently lack data to support this notion and thus we can only speculate. In line with this uncertainty and to avoid any confusions that this name might cause to readers the undifferentiated cells cluster was renamed as undefined and the text was modified so that to clearly show that at this point its identity can only be speculated. Moreover, an additional panel was added in Figure 1—figure supplement 1 (panel E) to show that the number of features and of molecules of this cluster is comparable to the ones detected in well-defined cell type families (eg. esophagus) suggesting that this cluster is real and not empty droplets. Note that a cluster of similar identity has been previously found in the study of Perillo et al., 2020 also reinforcing the fact that this is a real cell type family. Furthermore, an updated panel Figure 1—figure supplement 1 (panel F) showing the distribution and expression level of all the 69 marker genes of this cluster was added to show:

– The low number of marker genes that define this cell type.

– The relative low expression of them in that domain.

– That most of them are shared with ectodermally derived cell type families highlighting that this cluster could be of ectodermal origin.